# Productivity and Species Diversity of Plant Communities Are Higher inside than outside the West Ordos National Nature Reserve, Northern China

**DOI:** 10.3390/plants13050660

**Published:** 2024-02-27

**Authors:** Pu Guo, Qi Lu, Shuai Li

**Affiliations:** 1Institute of Ecological Conservation and Restoration, Chinese Academy of Forestry, Beijing 100091, China; 2Institute of Desertification Studies, Chinese Academy of Forestry, Beijing 100091, China; 3Experimental Center of Desert Forestry, Chinese Academy of Forestry, Dengkou 015200, China; lisshaxs@163.com

**Keywords:** desert steppe, aboveground biomass, plant diversity, nature reserve, soil nutrients

## Abstract

Nature reserves play an important role in the protection of biological habitats and the maintenance of biodiversity, but the performance and mechanisms of desert steppe nature reserves in improving plant community productivity, biodiversity and soil nutrient content are still largely unknown. To investigate the conservation effects of desert steppe nature reserve management on plant productivity and biodiversity, we compared the plant biomass, diversity and soil nutrient content inside and outside the West Ordos National Nature Reserve through sample survey, biomass determination, diversity index calculation and soil nutrient content determination. We found the following: (1) The aboveground biomass and belowground biomass of plant communities in the nature reserve were significantly larger than those outside the reserve; and the aboveground biomass of plant communities in shrub–steppe was significantly larger than that of herb grassland in both the nature reserve and the outside of the reserve. (2) The Margalef richness index, Shannon–Wiener index and Simpson index were significantly greater in the reserve than outside the nature reserve. In the desert steppe, the establishment of the nature reserve increased the α-diversity of the plant community. (3) The soil organic carbon (SOC) and soil total nitrogen (STN) were greater in the nature reserve than outside the reserve, and for the 10–20 cm and 20–40 cm soil layers, the SOC and STN were significantly greater in the core protected zone of the reserve than outside the reserve. The reserve significantly increased the nutrient content of the deeper soil layers. (4) The aboveground biomass of the plant community had a significant positive linear relationship with the species richness index, the Shannon index, and the Simpson index. There was a positive correlation between the diversity of the plant community and the soil nutrients. In summary, the nature reserve improved local plant productivity, biodiversity and the soil nutrient content, and the soil nutrient content in deeper soil layers may be the driving factor for the increase in productivity and biodiversity, which deepens our understanding of the conservation effectiveness of the nature reserve and its mechanisms.

## 1. Introduction

Our world is facing a crisis of rapid biodiversity loss, and biodiversity is crucial for maintaining ecosystem functions in a changing environment [1]. Species diversity is a fundamental ecological indicator of the diversity and complexity of plant communities and the ecosystem structure [2,3]. Species diversity can be quantified with various indices. Among them, α-diversity is a measure of the number and evenness of species distribution within a community, reflecting the ability of each species to adapt to the environment and utilize resources [4], as well as the abundance and evenness of species in the community and the stability of the community [5]. Beta diversity refers to the difference in species composition between different communities along an environmental gradient or the turnover rate of species along an environmental gradient, which can reflect the diversity of habitats in a community [6,7] and the interrelationship between environmental conditions and communities. Ecosystem productivity is an important indicator of ecosystem functioning and an important component of ecosystem services [8], and plant biomass is an important component of ecosystem productivity and an important indicator for assessing ecosystem productivity [9].

Soil is the environmental basis for the survival of terrestrial plants, and it plays an important role in the formation and development of the structure and function of plant communities; it is also an important environmental factor that affects the productivity and biodiversity of plant communities [10,11]. Plant communities and soil environments are interconnected and interact with each other through physical and chemical processes, which are important for maintaining the stability of the ecosystem structure [12]. The soil nutrient content affects the reproduction and development of plant community species and is an important driver for maintaining the sustainability of ecosystem functions [13,14,15]. Soil carbon, nitrogen and phosphorus are important elements in ecosystem material cycling and energy flow, and play an important role in regulating plant community structure and maintaining community function. Exploring the interrelationships between species diversity and soil factors is of great significance to the understanding of the mechanism of maintaining species diversity in plant communities as well as ecosystem management and conservation [16].

Nature reserves play an important role in biodiversity conservation. Nature reserves can also improve the ecosystem services and functions of the reserved area and ensure regional ecological security. Desert steppe is a kind of grassland in transition from grassland to desert, and it is also the driest type of grassland vegetation. The desert steppe in Inner Mongolia Autonomous Region is an important animal husbandry base and an important ecological security barrier in the north of China [17,18]. Although there are few desert nature reserves, each reserve has a large area, which is very important to maintain biodiversity and ecosystem services in Northern China [19]. The nature reserves of desert steppe are located in a more fragile environment, with problems of overgrazing, drought and land degradation, so it is crucial to strengthen the management and protection of the nature reserves of desert steppe. To assess the conservation effect of nature reserves, it is crucial to compare the differences between the ecological environment inside and outside the nature reserves [20]. However, many studies have focused on forest and wetland nature reserves [21,22], and relatively few studies have been conducted on the nature reserves of the desert ecosystem. Most of the studies are based on remote sensing methods [23], thus lacking ground investigation. West Ordos National Nature Reserve is a typical nature reserve of the desert ecosystem in North China, protecting ancient relict endangered plants and desert ecosystems such as *Tetraena mongolica* and *Helianthemum songaricum*. It is the largest national nature reserve containing desert grassland and holds high research value.

The objective of this study is to compare the plant community productivity, biodiversity, and soil nutrients inside and outside the West Ordos National Nature Reserve. Additionally, we aim to discuss the relationships between productivity and biodiversity, as well as biodiversity and soil nutrients. This study provides a scientific evaluation of the protection effect of desert nature reserves and suggestions for improving the management level of nature reserves.

## 2. Results

### 2.1. Comparison of Biomass of Plant Communities inside and outside Nature Reserve

The cover of herb grassland was the highest in the core protected zone of the nature reserve and the cover of herb grassland was the lowest outside the nature reserve (Figure 1A). The cover of shrub grassland and herb grassland in the nature reserve was significantly larger than that outside the reserve, in which the cover of herb grassland in the core reserve of the nature reserve was significantly larger than that in the general control zone; the cover of herb grassland in the core reserve of the nature reserve was significantly larger than that of shrub grassland, and there was no significant difference in the cover of shrub grassland and herb grassland in the general control zone of the reserve and outside the reserve. The average height of shrub grassland plant communities in the core protected zone of the nature reserve was the highest, and the average height of herb grassland communities outside the reserve was the lowest (Figure 1B). The average height of the plant community in the nature reserve was significantly greater than that outside the reserve; in addition, there was a significant difference in the average height of the plant community of shrub grassland and herb grassland in the core protected zone of the reserve and outside the reserve, which manifested itself as shrub grassland > herb grassland. The aboveground biomass of the shrub grassland plant community was the highest in the core protection area of the nature reserve, and the aboveground biomass of the herb grassland plant community was the lowest outside the reserve (Figure 1C). The aboveground biomass of the plant communities in the nature reserve was significantly larger than that outside the reserve; the aboveground biomass of the shrub grassland was significantly larger than that of the herb grassland, both inside and outside the reserve (Figure 1C). For the belowground biomass of plant communities, it was still significantly greater inside than outside the nature reserve; in the nature reserve, the belowground biomass of the plant communities in shrub grassland was significantly greater than that in herb grassland (Figure 1D) (*p* < 0.05).

### 2.2. Comparison of Plant Community Biodiversity inside and outside Nature Reserve

#### 2.2.1. Plant Community α-Diversity

The α-diversity indices of the plant communities were all within > outside the protected zones (Figure 2). The Margalef richness index, Simpson’s index, Shannon–Wiener’s index and Pielou’s evenness index were all significantly larger in the nature reserve than outside the protected zones (*p* < 0.05), while the α-diversity indices in the core protected zones and general control areas were not significantly different. The control areas were not significantly different in their α-diversity indices. In herb grassland, Simpson’s index and Shannon–Wiener’s index were significantly greater in the nature reserve than outside the nature reserve, and there was no significant difference between Simpson’s index and Shannon–Wiener’s index in the core protected zone of the nature reserve and the general controlled zone; the Pielou evenness indices in the core protected zone of the nature reserve, in the general controlled zone, and outside the reserve were not significantly different. There was no significant difference in the Margalef richness index between the core protected zones, general controlled zones and outside protected zones, with an order of core protected zones > general controlled zones > outside protected zones (*p* < 0.05). The α-diversity indices of the plant communities were not significantly different between shrub grassland and herb grassland. The above results indicated that in desert steppe, the establishment of a nature reserve significantly increased the α-diversity of the plant communities, and that the species richness of the core protected zones was also significantly higher than that of the general controlled zones in the nature reserve.

#### 2.2.2. Plant Community β-Diversity

The relationship matrix of the β-diversity index (Jaccard’s similarity coefficient) among the sample plots is shown in Table 1, and the range of the β-diversity index among the sample plots is from 0.125 to 0.385; this is a large fluctuation, indicating that the habitat differences among the three sample plots are relatively large. In terms of inside and outside the nature reserve, the β-diversity indices of shrub grassland in the core protected zone and outside the reserve were the largest, while the β-diversity indices of shrub grassland in the core protected zone and outside the reserve were the lowest. This shows that the habitat difference between the shrub grassland in the core protected zone and outside the reserve is small, while the habitat difference between the herb grassland in the core protected zone and outside the reserve is great. In terms of inside and outside the shrub, the β-diversity indices of the shrub grassland and herb grassland in the core reserve were the largest, while those of the shrub grassland and herb grassland outside the reserve were the smallest, suggesting that the establishment of the nature reserve reduced the habitat differences between shrub grassland and herb grassland.

### 2.3. Comparison of Soil Nutrient Contents inside and outside Nature Reserve

As shown in Table 2, both the SOC and STN were greater in the core protected zone than outside the protected zone, and for the 0–10 cm soil layer, there was no significant difference between the SOC and STN contents in the core protected zone, general controlled zone and outside the protected zone. For the 10–20 cm and 20–40 cm soil layers, the SOC and STN contents in the core protected zone of the nature reserve were significantly greater than those outside the protected zone. For shrub grassland, the SOC and STN contents in the 20–40 cm soil layer in the nature reserve were significantly greater than those outside the reserve. For herb grassland, there was no significant difference in the SAP content between the soil layers inside and outside the nature reserve. The SAP contents of the 0–10 cm and 10–20 cm soil layers of shrub grassland in the core protected zone were significantly larger than those outside the protected zone.

### 2.4. The Relationship between Plant Community Aboveground Biomass and α-Diversity

The relationship between the plant community aboveground biomass and community α-diversity is shown in Figure 3. In this study area, the plant community aboveground biomass had significant positive linear relationships with the Margalef richness index, Shannon–Wiener index and Simpson index, where biomass had the strongest correlation with the Shannon index, with R^2^ = 0.487; meanwhile, biomass had no significant correlation with the community Pielou index.

### 2.5. Correlation of Plant Community α-Diversity with Soil Nutrients

As shown in Figure 4, the Pearson correlation analysis showed that there were positive correlations between the plant community diversity and soil nutrients in all cases. Among them, the Margalef richness index and Shannon–Wiener index had a significant positive correlation with the SAP in the 0–10 cm soil layer; the Margalef richness index, Shannon–Wiener index and Simpson index had a significant positive correlation with the SOC content in the 10–20 cm soil layer; and all four diversity indices had a significant positive correlation with the 10–20 cm STN. The Margalef richness index and Shannon–Wiener index showed a significant positive correlation with the SAP in the 10–20 cm soil layer. All four diversity indices showed a significant positive correlation with the SOC in the 20–40 cm soil layer. The Margalef richness index, Shannon–Wiener index and Simpson index showed a significant positive correlation with the STN content in the 20–40 cm soil layer. The Shannon–Wiener index showed a significant positive correlation with the SAP in the 20–40 cm soil layer. The α-diversity index was more strongly correlated with the SOC and STN content in the 10–20 cm and 20–40 cm soil layers than in the 0–10 cm soil layer.

## 3. Discussion

### 3.1. Differences in Plant Community Characteristics and Soil Nutrients inside and outside Desert Steppe Nature Reserves

In this study, the cover and average height of the plant communities in the nature reserve were significantly larger than those outside the reserve, and the cover and average height of herb grassland in the core protected zone of the reserve were 5.4 and 2.9 times higher than those outside the reserve, respectively. This is attributed to the protection measures of the nature reserve, which effectively prevented the disturbance of grassland plants by grazing and other human activities, and promoted the recovery of grassland and plant growth and development, thus increasing the grass cover and the height of plant communities. Some studies have shown that the community cover of grassland is significantly improved after enclosure and nature reserve protection [24], and the present study is consistent with the results of the above studies. This study concluded that the aboveground biomass of plants in the nature reserve was significantly higher than that outside the reserve. Additionally, the aboveground biomass of herbaceous grassland plants in the core protected zone of the nature reserve increased by 1.5 times compared with that outside the reserve. This increase was attributed to the absence of anthropogenic disturbances. Activities such as grazing outside the reserve have been found to reduce plant biomass and affect the accumulation of matter and energy, ultimately impacting plant growth and reproduction. This results in decreased grassland productivity. However, the establishment of a nature reserve greatly reduces this situation and increases grassland productivity. This is consistent with the results of previous studies [25,26]. This study also found that the aboveground biomass of shrub grassland was significantly larger than that of herb grassland, both inside and outside the nature reserve, which was related to the protective effect of shrub [27], the “fertility island effect” [28,29] and the improvement of the microenvironment [30].

The species diversity of grassland communities plays an indispensable role in maintaining the structure and function of grassland ecosystems, as well as their sustainability [31]. This study found that the species richness index and Shannon’s index were significantly higher within the nature reserve compared to outside of it. Specifically, the species richness of herbaceous grassland in the core protected zone was significantly higher than that in the control zone and outside the reserve. These results suggest that strict closure and protection measures are more favorable for reducing habitat loss and the development of the plant soil seed bank, bringing the community closer to the local potential diversity level. These findings are consistent with those of previous studies [32,33,34,35]. This study also found that, except for species richness, there was no significant difference in α-diversity between the core protected zone of the nature reserve and the general control zone for either shrub grassland or herb grassland, which is not consistent with the moderate disturbance hypothesis. This may be related to the differences in environmental resources and environmental heterogeneity in this study area, or it may be due to the fact that the biomass of plant communities did not reach the threshold that drives the decline of species richness [36]. β-diversity refers to the difference in species composition between different communities along an environmental gradient or the rate of species turnover along an environmental gradient, which can reflect the diversity of the community’s habitats [6,7]. The results of this study showed that the β-diversity indices of the shrub grassland and herb grassland in the core protected area of the nature reserve were the largest, while the β-diversity indices of the shrub grassland and herb grassland outside the reserve were the smallest, which could be attributed to the fact that the competition between shrubs and herb grassland in the plant community outside the nature reserve was more intense, and that the exchange of material and energy was more frequent [37], which reduced the utilization efficiency of water and nutrients [38]; accordingly, the original plant species changed. In the nature reserve, the competition between shrubs and herbaceous plants is reduced, which provides a wider range of “ecological niches” for all kinds of plants, and thus reduces the difference in habitat between shrub grassland and herb grassland.

This study showed that for the 0–10 cm soil layer, there was no significant difference in the SOC and STN contents in the core protected zone, buffer area and outside the protected area of the nature reserve. For the 10–20 cm and 20–40 cm soil layers, the SOC and STN contents in the core protected zone of the reserve were significantly larger than those outside the reserve, and the SOC and STN contents in the 20–40 cm soil layer of the shrub grassland in the nature reserve were significantly larger than those outside the reserve. This may be due to the fact that the shrub root system of the shrub grassland in the protected area is more developed, and that the deeper layers of the soil contain more plant roots decomposed into organic matter immobilized in them [39].

### 3.2. The Relationship between Plant Community Aboveground Biomass and α-Diversity

It has been shown that the relationship between plant productivity and biodiversity has various forms, and that the relationship between productivity and biodiversity will be changed when there are changes in the material cycle and energy flow and heterogeneity of environmental resources [40]; it has also been shown that the relationship between productivity and biodiversity has the following forms: linear positive, linear negative, and single peak [41,42]. The results of this study showed that the aboveground biomass of plant communities had a significant positive linear relationship with the species richness index, the Shannon index and the Simpson index, in which the biomass had the strongest correlation with the Shannon index, R^2^ = 0.487, while the biomass did not have a significant correlation with the Pielou index of the community. This is consistent with the ecological niche theory, because in plant communities, there are differences in the ecological niche of different species, and the higher the species diversity, the wider the “ecological niche space” occupied by plant communities; this can improve the efficiency of plant communities regarding the utilization of environmental resources, and further increase the productivity of the community [43]. The aboveground biomass and species diversity of the plant communities in this study did not show a single-peaked relationship, which is different from the results of a previous study [44]. This may be because the biodiversity of the plant communities in this study area has not yet reached the threshold of decreasing productivity, or it may be due to the fact that the relationship between productivity and biodiversity in the study area is a simple linear positive correlation. In-depth studies on the relationship between the aboveground biomass and biodiversity of plants inside and outside nature reserves and the mechanism of their influence are needed. We need to collect more samples to improve further research.

### 3.3. Correlation of Plant Community α-Diversity with Soil Nutrients

It has been shown that higher biodiversity increases the amount of litter fall and the belowground biomass of plant communities, which further increases the soil nutrient content, and may also promote soil microbial diversity, which in turn leads to more rapid soil nutrient cycling [45]. In turn, soil nutrients may promote plant growth and development, further increasing plant community biodiversity. In this study, it was found that there was a positive correlation between both plant community diversity and soil nutrients. This is similar to the results of Xu’s study. In this study, we also found that Margalef’s richness index and Shannon–Wiener’s index were significantly and positively correlated with the SAP in the 10–20 cm soil layer, and all four diversity indices were significantly and positively correlated with the SOC in the 20–40 cm soil layer. The STN content in the 20–40 cm soil layer was significantly positively correlated, which indicated that the plant community diversity was more closely related to the nutrient content of the deeper soil layer. The nutrient content of the deeper soil layer in the nature reserve was significantly greater than that outside the reserve, which provided favorable conditions for improving the biodiversity of plant communities in the reserve, and to some extent explained the phenomenon that productivity and biodiversity were higher in the reserve than outside the reserve in this study.

## 4. Materials and Methods

### 4.1. Study Site

The research area is located in the West Ordos National Nature Reserve (106°40′–107°44′ E, 39°14′–40°11′ N, 1100–2000 m asl), Otog Banner, Inner Mongolia Autonomous Region. The West Ordos National Nature Reserve was established in 1995, and it has a total area of 436,116.40 hectares. The main species in this area are *Zygophyllum xanthoxylum*, *Tetraena mongolica*, *Helianthemum songaricum*, *Reaumuria songarica*, and *Stipa tianschanica*. The nature reserve belongs to the desert ecosystem nature reserve, and mainly protects the ancient remnants of endangered plants and desert ecosystems such as *Tetraena mongolica* and *Helianthemum songaricum.* The region has an arid and semi-arid continental climate in the mesotemperate zone. In this area, the winter is long and cold, while the summer is short and warm. Spring is characterized by strong winds, and the weather in fall is stable. The frost-free period in this area lasts for 129 days. The average annual temperature is 7 °C, with the lowest temperature recorded at −36.8 °C and the highest at 37 °C. The annual evaporation rate is 2470.4 mm, which is 9.1 times the amount of precipitation. The average wind speed is 3.4 m/s, with an average of 36 days of winds of grade 6 or above in a calendar year. Of these, 19 days (52.5%) occur in spring. The soil consists mainly of chestnut–calcium soil.

### 4.2. Sample Selection and Setting

Nature reserves are divided into core protected zones and general controlled zones. The core protected zones are primarily responsible for limiting human activities to the greatest extent possible in order to fulfill their protection function. The general controlled zones, on the other hand, are responsible for fulfilling their protection function while also considering public service functions such as scientific research, education, and recreation. In this research area, the core protected zone is closed and human activities are strictly prohibited. Necessary human activities, such as grazing, may be limited to the general control zone. Production activities outside the reserve are not restricted. The primary human activities in this study area are cattle and sheep grazing and crop planting. The field survey was carried out in August 2021. Three locations of the core protected zone, general controlled zone and outside the reserve were selected as the study sample plots, with two vegetation types: shrub grassland and herb grassland. “Herb grassland” denotes grassland with herbs as the dominant plant, while “shrub grassland” denotes grassland with shrubs as the dominant plant. To avoid the influence of topographic factors on the experimental results, the sample plots were selected in areas with similar geomorphology and flat terrain, and three quadrats (1 m × 1 m for herb grassland and 2 m × 2 m for shrub grassland; the data were converted to 1 m^2^ after measurement) were randomly set up in the grassland of the two planting types in each sample plot for the investigation of the plant communities, and three sample plots were located 50 m apart from each other.

### 4.3. Sample Collection and Determination Method

Indicators of the plant community survey include the species composition, density, height and cover of all plants in the sample plot. To determine the biomass of the herbaceous plants, all aboveground parts of the grass plants in the sample plot were cut and sorted by species, brought back to the laboratory and dried in an oven at 65 °C to a constant weight, and then weighted. For shrub plants, the standard plants whose sizes and heights were representative of the average state of the community’s thickets were first selected in the surveyed sample plot. The aboveground part of the standard plant was then cut with scissors near the ground. The samples were kept in paper bags and returned to the laboratory to determine their biomass using the drying method. The sample biomass was calculated according to the following formula: biomass of shrubs = biomass of standard plants x number of shrub plants. After plant sampling was completed, one 0–40 cm root sample was taken from each sample plot using a root drill that was 7 cm in diameter. The root samples were rinsed with water using a 2 mm sieve. The rinsed root samples were then dried in an oven at 65 °C until constant weight and weighed to calculate the belowground biomass of the plants. Soil samples were collected in synchronization with the plant community survey. The ‘S’-shaped five-point mixed sampling method was used to collect soil samples from each plot at depths of 0–10 cm, 10–20 cm, and 20–40 cm [46]. Three replicates were taken for each layer of the soil samples. The samples were preserved in a 4 °C ice box after collection and analyzed in the laboratory for nutrient content [47].

### 4.4. Statistical Analysis

The α-diversity indices [48] (Margalef’s richness index, Simpson’s index, Shannon–Wiener’s index and Pielou’s evenness) and β-diversity index (Jaccard’s index) [49] of the plant community were calculated for each sample site based on the field observation data. The analysis was conducted using SPSS 17.0 software. One-way ANOVA was used to analyze the alpha diversity index, coverage, average height, aboveground biomass, belowground biomass, and soil nutrients of the plant community in the core protected zone, general control zone, and outside the reserve. Multiple comparisons were performed using Tukey’s HSD. The significance level for differences was set at *p* = 0.05. Simple linear regression analysis was used to analyze the relationship between the aboveground biomass and plant community alpha diversity index. Additionally, the Pearson correlation coefficient was used to analyze the relationship between the plant community alpha diversity index and soil nutrients.

## 5. Conclusions

We conclude that, based on the ground survey data, in desert grasslands, the plant community productivity and biodiversity are significantly higher inside than outside nature reserves. This study also compared the plant belowground biomass inside and outside the nature reserve, and we conclude that the belowground biomass of plant communities in the reserves was also significantly higher than outside the reserves. The soil nutrient content was higher inside than outside the protected area, and there was a positive correlation between the biodiversity and soil nutrient content at all levels, and the correlation between the diversity index and the SOC and STN content of the 10–20 cm and 20–40 cm soil layers was stronger compared to that of the 0–10 soil layers, which to some extent explains the phenomenon that the productivity and biodiversity were higher in the protected area than outside the protected area in the present study. All these results help us to better understand the mechanism of the protective effectiveness of nature reserves on local plant communities and soils. To enhance the productivity of local plant communities, species diversity, and soil nutrients, it is essential to improve the management of nature reserves.

## Figures and Tables

**Figure 1 plants-13-00660-f001:**
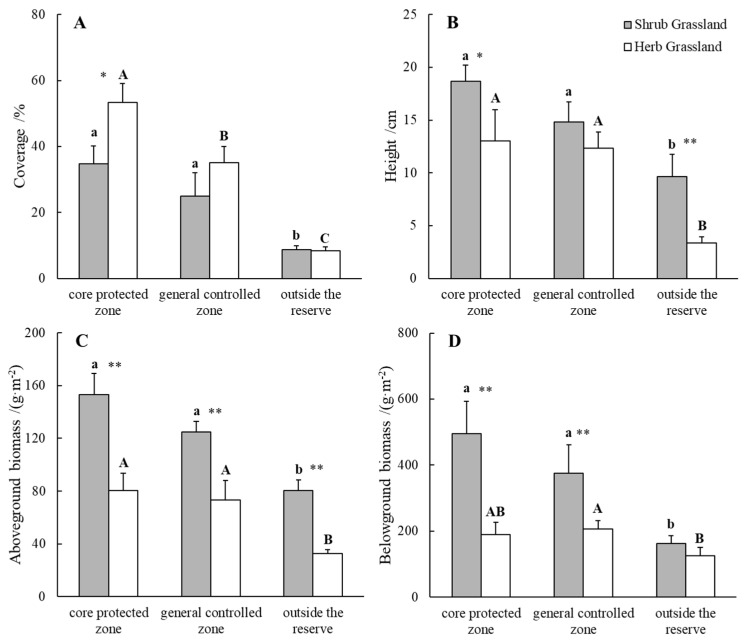
The plant community coverage (**A**), average height (**B**), aboveground biomass (**C**) and belowground biomass (**D**) inside and outside nature reserve. Note: Different lowercase letters indicate significant differences between locations of shrub grassland (*p* < 0.05); different uppercase letters indicate significant differences between locations of herb grassland (*p* < 0.05). “*” indicates that the differences between different plant communities at the same location are significant at the *p* < 0.05 level, and “**” indicates the *p* < 0.01 level of significance.

**Figure 2 plants-13-00660-f002:**
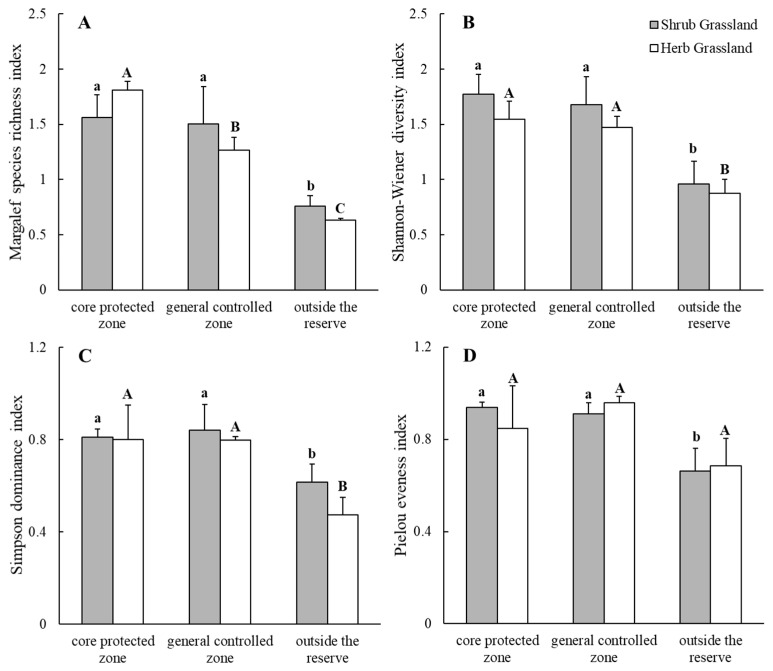
Plant community Margalef species richness index (**A**), Shannon–Wiener diversity index (**B**), Simpson dominance index (**C**) and Pielou eveness index (**D**) inside and outside the nature reserve. Note: Different lowercase letters indicate significant differences between locations of shrub grassland (*p* < 0.05); different uppercase letters indicate significant differences between locations of herb grassland (*p* < 0.05).

**Figure 3 plants-13-00660-f003:**
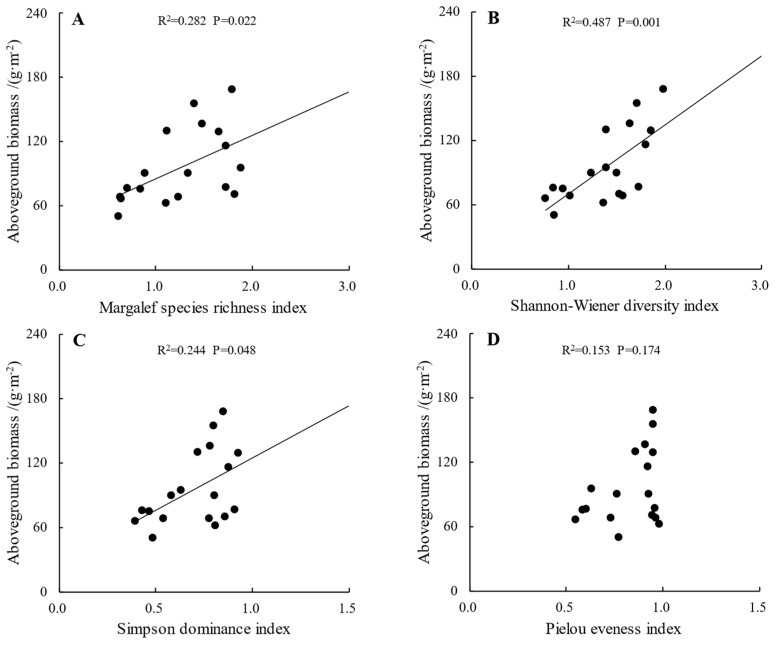
The relationship between plant community aboveground biomass and Margalef species richness index (**A**), Shannon–Wiener diversity index (**B**), Simpson dominance index (**C**) and Pielou eveness index (**D**).

**Figure 4 plants-13-00660-f004:**
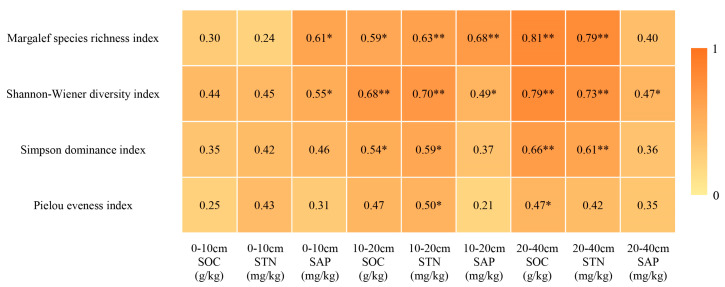
The relationship between plant community aboveground biomass and α-diversity. Note: “*” indicates significant correlation (*p* < 0.05), “**” indicates significant correlation highly significant (*p* < 0.01). SOC: Soil organic carbon; STN: Soil total nitrogen; SAP: Soil active phosphorus.

**Table 1 plants-13-00660-t001:** Relationship matrix of β-diversity index inside and outside the nature reserve.

Plots		Core Protected Zone		General Controlled Zone		Outside the Reserve	
	Vegetation Types	Shrub Grassland	Herb Grassland	Shrub Grassland	Herb Grassland	Shrub Grassland	Herb Grassland
Core protected zone	Shrub Grassland	1	0.333	0.231	0.167	0.385	0.308
	Herb Grassland		1	0.231	0.313	0.200	0.214
General controlled zone	Shrub Grassland			1	0.214	0.273	0.300
	Herb Grassland				1	0.267	0.125
Outside the reserve	Shrub Grassland					1	0.154
	Herb Grassland						1

**Table 2 plants-13-00660-t002:** Comparison of soil nutrient contents inside and outside the nature reserves.

Vegetation Types	Soil Depth	Plots	SOC/g·kg^−1^	STN/mg·kg^−1^	SAP/mg·kg^−1^
Shrub Grassland	0–10 cm	Core protected zone	4.70 ± 0.63	0.56 ± 0.11	4.79 ± 0.20 a
		General controlled zone	4.50 ± 0.62	0.48 ± 0.08	3.84 ± 0.89 ab
		Outside the reserve	4.05 ± 1.06	0.42 ± 0.07	2.52 ± 0.63 b
	10–20 cm	Core protected zone	6.83 ± 1.91 a	0.76 ± 0.21 a	3.57 ± 0.56 a
		General controlled zone	4.94 ± 0.44 ab	0.59 ± 0.04 ab	2.56 ± 0.60 ab
		Outside the reserve	3.04 ± 0.41 b	0.38 ± 0.04 b	1.71 ± 0.07 b
	20–40 cm	Core protected zone	4.90 ± 0.51 a	0.55 ± 0.01 a	3.36 ± 1.19
		General controlled zone	4.32 ± 0.50 a	0.51 ± 0.08 a	2.09 ± 0.39
		Outside the reserve	2.42 ± 0.22 b	0.29 ± 0.04 b	1.55 ± 0.19
Herb Grassland	0–10 cm	Core protected zone	3.08 ± 0.45	0.33 ± 0.03	4.20 ± 0.90
		General controlled zone	4.00 ± 1.40	0.51 ± 0.16	2.82 ± 1.11
		Outside the reserve	2.84 ± 0.11	0.32 ± 0.06	3.17 ± 0.75
	10–20 cm	Core protected zone	3.95 ± 0.96 a	0.52 ± 0.10 a	3.31 ± 1.17
		General controlled zone	3.76 ± 0.52 ab	0.47 ± 0.08 ab	2.03 ± 0.14
		Outside the reserve	2.20 ± 0.48 b	0.26 ± 0.07 b	2.05 ± 0.57
	20–40 cm	Core protected zone	4.24 ± 1.05 a	0.52 ± 0.13 a	2.03 ± 0.14
		General controlled zone	2.60 ± 0.26 b	0.32 ± 0.06 ab	1.83 ± 0.38
		Outside the reserve	1.86 ± 0.26 b	0.25 ± 0.04 b	1.75 ± 0.21

Note: Different lowercase letters represent the significant differences in different plots in the same soil layers and at the same vegetation types (*p* < 0.05). SOC: Soil organic carbon; STN: Soil total nitrogen; SAP: Soil active phosphorus.

## Data Availability

Data are contained within the article.

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
