# Peer review of "Productivity and Species Diversity of Plant Communities Are Higher inside than outside the West Ordos National Nature Reserve, Northern China"

_plants, 2024, doi:10.3390/plants13050660_

Round 1

Reviewer 1 Report

Comments and Suggestions for Authors

Comments on the Quality of English Language

The English is fine with the exception of the points noted in the review.  

Author Response

Dear reviewer:

Thank you very much for your review. Please see the attachment.

Reviewer 2 Report

Comments and Suggestions for Authors

Dear Authors,

This abstract outlines a study conducted to assess the conservation effects of desert steppe nature reserve management on plant productivity, biodiversity, and soil nutrient content. The researchers compared these factors inside and outside the West Ordos National Nature Reserve through sample surveys, biomass determination, diversity index calculations, and soil nutrient content determination. The research concludes that the productivity and biodiversity of the vegetation is higher within the nature conservation area, and the nutrient content of the soil is also more favorable. Making critical statements, the texts are generally well structured and valuable in terms of content, however, shortcomings can also be pointed out. For example, a more detailed analysis of the results and a more detailed description of the methods could improve the scientific content of the study. In addition, some sentences contain structural and grammatical errors that could be corrected by languange editing. Overall, it would be worthwhile to review and supplement the texts in the mentioned areas in order to get a more accurate and complete picture of the results and conclusions of the research.

Introduction:The Introduction is basically well structured, but could be enriched with some additional background information in order to better understand the context of the research.A more precise definition of the purpose and hypotheses of the research is lacking.

A large amount of data is presented in the Results section, but it is sometimes difficult to interpret or put it into context. It would be better if the data were analyzed and interpreted in more detail. Some results are not clearly related to the objectives of the research.

The results are analyzed in detail in the Discussion section, but sometimes they lack a broader context.Further comparisons could be made with the results of other, similar researches in order to better understand the significance of the results.

The methods are well described, but some details are missing, such as precise data collection methods or laboratory analytical procedures. It is not entirely clear from the details of the methods how the reliability and validity of the tests were guaranteed.

The Conclusion summarizes the main results, but some additional conclusions and recommendations would be needed to illuminate the next steps of the research and the future directions of the research field. The conclusions are not always linked back to the research objectives stated in the introduction.

Comments on the Quality of English Language

Some sentences contain structural and grammatical errors that could be corrected by languange editing.

Author Response

(The authors gave the same response as above.)

Reviewer 3 Report

Comments and Suggestions for Authors

An interesting paper which can be easily improved.

L153. This sentence is overly long and confusing to read. It should be broken up for ease of understanding.  I wonder if some should refer to “herb grassland”. Why is it referred to as “scrub” grassland?

L175 please spell out what SAP stands for (presumably soil available phosphorus).

L203, What does “relationship” mean here?

L215 the sentence beginning here is essentially the same as the sentence beginning L219.

L225. Another overlong sentence that should be broken down to several sentences.

L333-335. A redundant sentence, which has obviously been corrected in the succeeding sentence.

L343. Some more description of the characteristics of the different zones would help here. What are the principal herbivores in each zone and how are they controlled? Are the different zones fenced, to what height and for how long at the time of sampling? A brief summary of the previous paper would be helpful.

L371 to 373. A redundant part sentence, contained in the succeeding sentence with a reference.

Comments on the Quality of English Language

The expression is understandable but some sentences are overlong and can be sub-divided to increase readability. 

Author Response

(The authors gave the same response as above.)
